# World Models Should Prioritize the Unification of Physical and Social Dynamics

**Xiaoyuan Zhang**[1,2,3,*]   **Chengdong Ma**[1,3,*]   **Yizhe Huang**[1,2]   **Weidong Huang**[2],
**Siyuan Qi**[2],   **Song-Chun Zhu**[2,1,3,†]   **Xue Feng**[2,†]   **Yaodong Yang**[1,3,†]

[1] Institute for Artificial Intelligence, Peking University
[2] State Key Laboratory of General Artificial Intelligence, BIGAI, Beijing, China
[3] State Key Laboratory of General Artificial Intelligence, Peking University, Beijing, China

## Abstract

World models, which explicitly learn environmental dynamics to lay the foundation for planning, reasoning, and decision-making, are rapidly advancing in predicting both physical dynamics and aspects of social behavior, yet predominantly in separate silos. This division results in a systemic failure to model the crucial interplay between physical environments and social constructs, rendering current models fundamentally incapable of adequately addressing the true complexity of real-world systems where physical and social realities are inextricably intertwined. This position paper argues that the systematic, bidirectional unification of physical and social predictive capabilities is the next crucial frontier for world model development. We contend that comprehensive world models must holistically integrate objective physical laws with the subjective, evolving, and context-dependent nature of social dynamics. Such unification is paramount for AI to robustly navigate complex real-world challenges and achieve more generalizable intelligence. This paper substantiates this imperative by analyzing core impediments to integration, proposing foundational guiding principles (ACE Principles), and outlining a conceptual framework alongside a research roadmap towards truly holistic world models.

## 1  Introduction

The cognitive capacity of intelligent agents to construct and utilize internal "world models" for prediction, planning, and adaptive response [16, 59] represents a foundational principle of intelligence and serves as a significant paradigm for advancing artificial intelligence (AI). The development of AI world models, which endeavor to explicitly learn and predict environmental dynamics to underpin agentive planning, reasoning, and decision-making processes, is currently characterized by a period of dynamic and transformative expansion. Noteworthy advancements include the exploration of Large Language Models (LLMs) as nascent simulators of physical phenomena and as cognitive architectures for agents operating within simplified or text-centric environments [74]. Currently, sophisticated video generation models, such as Stable Video Diffusion [10], are achieving remarkable fidelity in predicting and synthesizing complex visual and, by extension, implicit physical dynamics. Furthermore, model-based reinforcement learning (MBRL) agents, exemplified by systems like DreamerV3 [36], have surpassed human performance benchmarks in complex interactive domains through the learning and utilization of internal dynamic representations of physical environments. These collective successes underscore a rapidly maturing capability to model discrete facets of our world with increasing precision and utility.

---

*Equal contribution
†Equal corresponding authors. Project website: https://sites.google.com/view/world-model-position

39th Conference on Neural Information Processing Systems (NeurIPS 2025) Position Paper Track.

The long-term aspiration of world models is to predict the multifaceted complexities of the real world. As illustrated in Figure 1, such complexity inherently encompasses both **physical dimension**, governed by natural laws (e.g., gravity), and **social dimensions**, arising from agentive interactions, subjective beliefs of states, and collective behaviors (e.g., human emotions, social relationships). These two categories of prediction, while fundamentally different and often requiring distinct learning approaches, are inextricably linked in any veridical representation of reality. This paper, therefore, approaches existing world model research through the crucial lens of this physical-social duality, aiming to facilitate more holistic future development.

However, this burgeoning progress in modeling specific dimensions often obscures several profound key problems that constrain the aspiration for truly comprehensive and generalizable world understanding. A predominant limitation is the systemic inadequacy in modeling the rich, bidirectional interplay between the physical environment and the intricate social fabric woven by its intelligent inhabitants. This prevalent separation of physical and social modeling, where world models are often confined to a single dimension of reality, exposes a fundamental incompleteness.

How, then, can these deep-seated challenges be surmounted? **Our Position: The next significant leap in AI world model development must be defined by, and will critically depend upon, the deep and bidirectional unification of physical and social predictive capabilities.** We assert that a truly successful, general-purpose world model must holistically integrate its understanding and predictive capacity for both domains. Here, physical dynamics prediction pertains to forecasting objective material states and transformations governed by natural laws. Social dynamics prediction involves anticipating behaviors, internal cognitive-affective states, and collective patterns of intelligent agents. Their unification demands modeling their profound interdependencies and reciprocal causal influences—how social intent shapes physical action, how physical context constrains social possibility, and how this feedback loop drives their co-evolution.

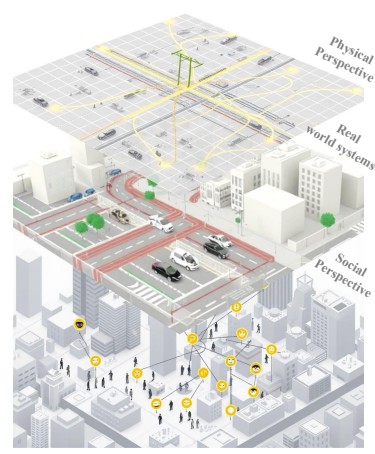

Figure 1: **Real-world systems are composed of both physical and social dimensions.** Physical aspects include vehicle movement, pedestrian flows, and power grid distribution (lines), while social aspects encompass competitive/cooperative relationships (connecting lines) and emotional states (facial expressions).

Consider common scenarios where this integration is paramount. Predicting urban traffic flow reliably fails if models only address vehicle kinematics (physical) without accounting for driver stress or adherence to social norms (social), which dramatically alter physical patterns. Similarly, effective human-robot collaboration necessitates modeling not just physical assembly but also the social dynamics of trust and communication. Without such integration, models offer a fractured view, unable to explain or predict these complex physical-social phenomena, thereby failing to resolve the aforementioned key problems. The current divergence of physical and social world modeling stems from formidable impediments: the representational chasm between objective physical data and subjective social constructs; the complexity of their entangled, bidirectional dynamics; the scarcity of rich, co-registered data; and the challenges in robustly evaluating integrated models.

To navigate these obstacles, this paper, drawing inspiration from cognitive science, sociology, and systems theory, proposes a principled approach. We systematically organize and analyze existing approaches to world modeling through dual lenses of physical and social dimensions in section 2. Building upon this foundation, we elucidate the inextricable linkage between social and physical dynamics, establishing the fundamental ACE principles to guide the study of world models. This culminates in the proposition of a conceptual framework and research roadmap aimed at developing holistic world models that bridge the physical-social divide in section 3. Our formulation paves the way for constructing more robust and socially-aware artificial intelligence systems through integrated modeling of multi-agent intentionality, socio-physical constraints, and emergent behavioral patterns.

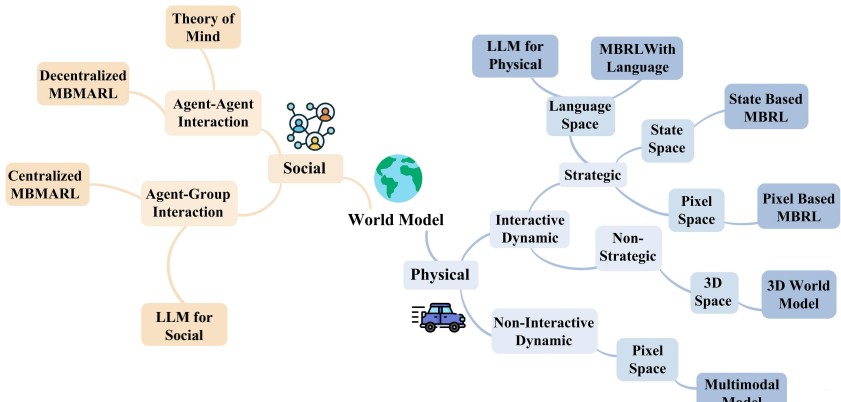

Figure 2: **Physical & Social world model diagram summarizing.** The social aspect is divided into Agent-Agent and Agent-Group Interaction, while the physical aspect distinguishes between interactive and non-interactive processes. Interactive dynamics are further classified into strategic and non-strategic interactions. Modalities are categorized into language, state, pixel, and 3D spaces. This is not a strict classification but a representative summary of current research directions. Table 1 and Table 2 provide detailed examples.

## 2 The Duality of World Model Predictions: Physical and Social Dimensions

The rapid proliferation and increasing sophistication of AI world models, aimed at learning environmental dynamics for prediction and planning [16, 59], underscore their pivotal role in advancing intelligent systems. However, to truly navigate the full complexity of real-world systems, where objective physical laws are inextricably intertwined with subjective human behaviors and evolving societal structures, it is imperative to critically examine not only what current models predict, but also what they overlook or fail to capture adequately. This section distinguishes world model predictions along two fundamental axes: the **physical dimension**, concerned with material reality and natural laws, and the **social dimension**, focused on agentive interactions, subjective beliefs of states, and collective dynamics. This distinction is not arbitrary. It reflects deep-seated differences in the nature of the phenomena being modeled, the governing regularities, and ultimately, the methodologies required for effective prediction. By surveying the current landscape through this dual lens, we aim to highlight not only domain-specific strengths but, more importantly for our position, the systemic limitations arising from their prevalent separation, thereby underscoring the pressing need for their unification.

### 2.1 The Physical Dimension: Modeling Objective Reality and Natural Laws

World models targeting the physical dimension endeavor to capture the objective, material aspects of environments and entities, governed by discoverable natural laws. Their predictive efficacy hinges on accurately representing and simulating the evolution of physical states over time. Our conceptualization of this domain, visually summarized in Figure 2, distinguishes modeling approaches based on the nature of dynamics (interactive vs. non-interactive) and the primary data modalities employed (e.g., language, state, pixel, 3D). This framework helps to navigate the diverse methodologies developed for physical world modeling.

**Classification and Scope of Physical Predictions.** The prediction of physical dimension involves understanding several key aspects. As depicted in Figure 2, a primary distinction is between **interactive dynamics**, where an agent's actions directly influence the environment (central to MBRL, e.g., policy-based strategic interactions in TD-MPC [38]), and **non-interactive dynamics**, which involve predicting passively evolving systems (common in video generation from static inputs, e.g., DynamiCrafter [121]). Interactive dynamics are further classified into strategic and non-strategic interactions. Physical predictions also target either explicit physical quantities with clear semantic meaning (e.g., velocity and mass in MBPO [49]) or latent physical representations learned from high-dimensional sensory data (e.g., Dreamer series [34, 36]). Finally, these predictions are made

across diverse **modalities**, including language descriptions, structured state spaces, raw pixel data, and explicit 3D geometric representations (e.g., 3D-LLM [44], OccWorld [133]).

**Prominent Methodologies in Physical World Modeling:**   Intuitive Physics Models, inspired by human cognition [6, 16], such as MAC networks [48] or DCL [15], aim to extract explicit physical quantities from visual data and acquire commonsense understanding of physical principles. While these methods have demonstrated progress in structured reasoning, robust generalization to complex real-world scenarios remains a significant challenge.

Model-Based Reinforcement Learning (MBRL) agents construct models of environment dynamics for sample efficiency and planning [105, 19, 102, 33, 71, 80, 126]. Latent variable models like Dreamer [34, 35, 36] and DayDreamer [115] enable learning through "imagination". Transformer-based architectures like IRIS [69] show robust performance in real-robot control [72]. MBRL has mastered complex games (e.g., MuZero [96], Atari [52]). However, model error accumulation, generalization, and agent social complexities beyond simple game-theoretic interactions remain limitations.

Video Generation Models (VWMs), such as Stable Video Diffusion [10] and generative transformers (like Open-Sora [63, 75]), synthesize photorealistic videos, implicitly capturing complex physics. Aligning these for planning (e.g., VADER [81], acting from actionless videos [55]) and ensuring long-term consistency has emerged as a prominent research direction, driven by their unparalleled capacity for visual fidelity and physical property modeling. However, current architectures lack explicit mechanisms to model agents' social contexts or intentional states within generated scenarios, constraining their ability to reason about interactive dynamics in socially situated environments. LLMs for physical reasoning show emerging capabilities in qualitative reasoning about physical laws from text [39]. Approaches like WorldCoder [107] use LLMs to generate simulation code or plans. However, they lack direct perceptual grounding and modeling of embodied social interactions within physical contexts. 3D World Models focus on explicit, geometrically rich representations (e.g., NeRFs, 3D occupancy grids from OccWorld [133], 3D-LLM [44]) for detailed spatial reasoning. Computational cost and real-time dynamic updates are ongoing challenges. For a comprehensive list of related papers and detailed methodologies in physical world modeling, including additional examples and classifications, refer to Table 1 in Appendix B.

**Predominant Limitation of Current Physical World Models.**   Despite these remarkable advancements, a unifying limitation is their often superficial, or entirely absent, representation of the social agents and the complex social dynamics that unfold within these physical environments. When agents are incorporated, they are frequently modeled as simple reactive entities, or their behavior is prescribed by predefined policies or learned via reward functions that lack rich social contextualization. The intricate internal cognitive and affective states, and the dynamic social interactions that profoundly govern human (and increasingly, sophisticated AI agent) behavior in the physical world, are typically not primary modeling targets. This fundamental oversight means that while these models can impressively predict *how* a physical system might evolve under certain given actions, they critically struggle to predict *what* actions an intelligent, socially-situated agent will actually choose to take, *why* they take those actions, or how a *group* of such agents will collectively influence the physical world. This fundamentally constrains their applicability and reliability in a vast array of complex, human-centric real-world scenarios.

### 2.2   The Social Dimension: Modeling Subjectivity, Interaction, and Group Dynamics

The social dimension of world models addresses the inherently subjective, context-dependent, and evolving nature of individual behavior, relationships between agents, and collective phenomena. Drawing from foundational theories in sociology and psychology [68, 8], we conceptualize social quantities and their prediction at distinct yet interacting levels.

**Levels of Social Abstraction.**   Modeling social reality computationally involves a hierarchical view. The Individual Level pertains to an agent's internal cognitive and affective architecture: beliefs, intentions, goals, emotions, values, preferences, and personality traits [86]. As shown in Figure 2, the **Interaction Level** (Agent-Agent & Agent-Group) focuses on the dynamics between agents, such as communication, the evolution of social relationships (e.g., trust, power), and strategic or game-theoretic encounters. The Group Level encompasses emergent collective phenomena: social

norms, collective action, and cultural values. These levels provide a framework for categorizing and understanding different approaches to social modeling.

**Prominent Methodologies in Social World Modeling:**   AI Theory of Mind (ToM) and Mental State Inference systems, such as ToMnet [86] or M$^3$RL [100], explicitly attempt to model an agent's capacity to infer the unobservable mental states (beliefs, desires, intentions, emotions) of others. This is crucial for predicting nuanced social behavior, e.g., in Sally-Anne tests or strategic games like Stag Hunt [101, 46]. These models excel at representing aspects of social cognition and predicting behavior in socially strategic situations. Nevertheless, they are typically evaluated in simplified, often discrete, environments with limited physical complexity. Scaling robust ToM to open-ended, richly contextualized physical scenarios, and grounding inferred mental states in continuous physical interactions, remains a significant hurdle.

Model-Based Multi-Agent Reinforcement Learning (MBMARL), surveyed in [114], investigates how agents learn predictive models of their environment and each other's policies to improve coordination (e.g., CACC [66]) and competition [117]. Modeling rich social states beyond policies, learning effective communication (e.g., MACI [82]). However, the "world" in MBMARL is often an abstract game state or a simplified representation, not a rich, dynamic physical environment with its own immutable laws and complex affordances that co-shape social strategies.

LLMs serve as powerful foundations for social world models, enabling the prediction of subjective social dynamics, such as preferences and agent behaviors. They excel at modeling rich social dialogues and complex interaction sequences, generating human-like language and diverse social behaviors in textual or simplified settings (e.g., Generative Agents [74]). These systems effectively function as dynamic world models for predicting emergent social states. However, a shared limitation across these approaches is their operation in abstract or disembodied contexts, often lacking explicit modeling of how social predictions translate into, or are constrained by, physical actions, environmental affordances, or real-time sensory inputs. This significantly hinders their ability to capture nuanced physical-social entanglements. For a comprehensive list of related papers and detailed methodologies in social world modeling, refer to Table 2 in Appendix B.

**Predominant Limitation of Current Social World Modeling Efforts.**   While promising strides are being made in modeling specific social facets, from preferences to ToM and LLM-driven interactions, two overarching limitations persist from a unification perspective. Firstly, these efforts often occur in **abstracted or disembodied physical environments**, neglecting the crucial grounding and reciprocal influence of material reality on social dynamics. Secondly, even as standalone endeavors, dedicated research into "Social World Models" as a cohesive field, with the systemic depth seen in physical world modeling, remains underdeveloped. There's often an **insufficient focus on truly complex, multi-level social abstractions** (e.g., enduring norms, cultural dynamics, power structures) and a lack of unified theoretical underpinnings or standardized evaluation paradigms for social world modeling itself. This dual challenge—the internal complexities of comprehensive social modeling compounded by its detachment from robust physical grounding—severely restricts current capabilities in representing real-world socio-technical systems.

Examining the landscape of world models, which includes established physical prediction methods (Intuitive Physics, MBRL, VWMs, 3D Models) and emerging social simulators (ToM, MBMARL, LLM for Social) as summarized inTable 1 and Table 2, a significant imbalance and separation become evident. There is a clear underinvestment in comprehensive Social World Model development compared to its physical counterpart. Furthermore, and most crucially for our thesis, these two vital predictive dimensions are almost universally **modeled as distinct and separate endeavors**, with minimal attempts at deep, bidirectional integration.

**Our survey of physical and social world modeling paradigms reveals a critical juncture.** While physical models excel at objective dynamics, they often neglect the social agency driving real-world actions. Conversely, emerging social models, though capturing nuanced interactions, typically operate in abstracted physical contexts, lacking robust grounding and an understanding of reciprocal physical influence. This analysis yields two clear conclusions: firstly, a **systemic underdevelopment of comprehensive Social World Models** capable of handling complex social abstractions and evolving norms. Secondly, and more critically, a **profound lack of deep, bidirectional integration between current physical and social modeling efforts**. This "integration gap" is a fundamental barrier, not a mere missing feature. Consequently, neither purely physical nor purely social world models, in

their prevalent isolated forms, can adequately capture the dynamics of real-world systems where physical laws and social agency are inextricably entangled. Predicting complex phenomena, from societal adaptation to climate change, to the socio-technical impact of new technologies, or nuanced human-robot collaboration, demands an integrated understanding that current siloed approaches fail to provide. These challenges directly highlight the "key problems" (e.g., in robust prediction, causal reasoning, and multi-agent decision-making) that persist due to this lack of fusion. Therefore, our central position is unequivocal: to build AI systems capable of true comprehension and effective interaction in our multifaceted world, the systematic unification of physical and social predictive capabilities within world models is an urgent scientific and engineering imperative.

## 3 Integrating Physical and Social World Model

This section lays the foundational groundwork for achieving truly unified physical-social world models, moving beyond isolated approaches. We first delve into the profound and reciprocal interdependence of physical and social dynamics, articulating why an integrated understanding is indispensable not only for predicting the physical world through a social lens, but equally for grounding social realities within their material context. Building on this imperative, we propose a set of guiding principles (the ACE Principles) to navigate the complexities of this endeavor. Finally, we present a conceptual blueprint outlining the core components and interactions of an integrated physical-social world model and its broad applicability.

### 3.1 The Inextricable Link Between Physical and Social World

The aspiration to create world models that truly mirror reality compels a departure from paradigms predominantly focused on isolated physical or social predictions. As strikingly illustrated in real-world systems (see Figure 1), the physical and social dimensions are not merely parallel but are inextricably linked through continuous, bidirectional influence [51, 78, 76, 70]. Understanding this entanglement is paramount, as the limitations of current world models, highlighted in section 2, largely stem from neglecting or inadequately modeling these profound interdependencies. This subsection delineates key facets of this indispensable interplay, first examining how social dynamics shape physical reality, and then how physical contexts sculpt social phenomena.

**Social Shaping of Physical Reality.** The physical world, particularly where human agency is salient, is profoundly shaped by multi-scalar social forces. Firstly, at the agent level, social cognition—encompassing goals, beliefs, intentions, and emotions—acts as the engine of purposeful physical action [16]. A purely physical model might predict a ball's trajectory if thrown, but cannot explain the social intent (e.g., play, aggression) dictating the throw itself and its physical characteristics. In multi-agent contexts like urban traffic, vehicle dynamics are orchestrated less by pure mechanics and more by driver objectives, inferred social understanding, and adherence to norms (e.g., traffic laws), rendering purely physical long-term prediction untenable. Secondly, collective social forces and established structures actively sculpt our physical environment and its utilization [76, 70]. Urban landscapes and large-scale ecological changes are material manifestations of societal planning, economic systems, and cultural values. Accurately forecasting long-term environmental evolution thus necessitates modeling these potent societal drivers. Thirdly, social norms and relational structures function as an implicit rulebook for physical interactions, defining permissible actions and shaping the "social physics" of an environment, from pedestrian flows to teamwork coordination. World models must integrate these social rule systems for predictions to be both physically plausible and socially coherent.

**Physical Influence on Social Dynamics.** Conversely, the physical world is not a passive stage but an active constituent that constrains, enables, and profoundly shapes social phenomena. The **environment's physical affordances and constraints** (e.g., geography, resource distribution, technological artifacts) directly influence the range of possible social interactions, economic activities, and even the structure of societies. For example, resource scarcity can significantly alter social cooperative norms or incite conflict. Moreover, significant physical events—natural or human-induced—often act as potent catalysts for social change and adaptation. A natural disaster can reconfigure community bonds and decision-making processes, while a technological breakthrough can reshape communication patterns and social hierarchies. Current social models, frequently detached from rich, dynamic physical grounding (as noted in subsection 2.2), struggle to capture these crucial physical-to-social causal

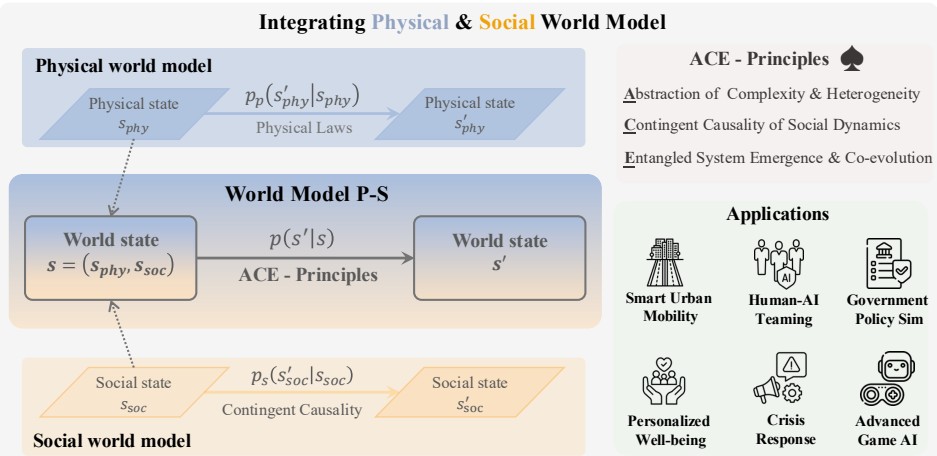

Figure 3: Integrated Physical-Social World Model. The physical state evolution (governed by physical laws) and social state evolution (driven by Contingent Causality) are unified into a physical-social world model, which follows the ACE Principles, leading to impactful applications.

pathways. Fundamentally, physical perception forms the bedrock of social understanding; agents infer others' intentions, emotions, and beliefs largely through observing their physical manifestations (expressions, gestures, actions) within a shared material context. Grounding abstract social concepts in concrete physical percepts and interactions is thus essential for any robust social reasoning.

**The Indispensable Entangled Loop and Its Implications.** This continuous, recursive feedback loop—where social agency systematically alters physical states, and physical realities dynamically modulate social cognition and interaction, is thus the defining characteristic of complex real-world systems populated by intelligent agents. Attempting to model either the physical or social dimension in isolation, or with only superficial linkages, inevitably leads to a fractured, incomplete, and ultimately inadequate understanding of reality. The "key problems" that plague current world model systems in achieving robust long-term prediction, deep causal reasoning, or effective multi-agent coordination in novel situations, are often direct consequences of failing to capture this deep **physical-social entanglement**. Therefore, constructing world models that genuinely reflect the richness and interconnectedness of our world necessitates integrated frameworks that explicitly model these foundational, bidirectionally influential socio-physical dynamics. This understanding forms the bedrock upon which our proposed guiding principles and conceptual framework are built.

## 3.2 Guiding Principles for Integrated Physical-Social World Models

To effectively navigate the profound complexities inherent in unifying physical and social world models, and to chart a course towards robust, insightful, and ethically-grounded integrated systems, we propose the **ACE Principles**. These three foundational tenets—Principled **A**bstraction of Social Complexity and Heterogeneity, Capturing the **C**ontingent Causality of Social Dynamics, and Enabling **E**ntangled System Co-evolution and Emergence—are specifically formulated to address the unique challenges posed by the deep entanglement of objective physical dynamics and subjective, evolving social constructs. They offer a coherent intellectual framework for the next generation of world model research and development.

**Abstraction of Social Complexity and Heterogeneity.** A primary challenge in integrating social dynamics lies in the inherent nature of social quantities. Unlike physical variables, social constructs (e.g., beliefs, norms, trust) are typically dimensionless, exceptionally heterogeneous, and operate within vast, often ill-defined conceptual spaces, varying significantly across individuals and contexts. Therefore, this principle advocates for **multi-level abstraction mechanisms specifically tailored for this social complexity and heterogeneity**. This necessitates models capable of: (a) representing diverse social information at various granularities, from transient individual mental states to enduring societal value systems; (b) effectively managing the profound heterogeneity within these representations; and (c) composing these social abstractions to inform decision-making and predict collective phenomena, while ensuring a meaningful interface with physical world representations.

**Contingent Causality of Social Dynamics.** A defining distinction from physical systems, governed by largely immutable and universal causal laws, is that the causal relationships patterning social dynamics are inherently **contingent: they are mutable over time, highly sensitive to specific socio-physical contexts, and significantly shaped by agentive interpretation and strategic construction** [24]. While a physical law dictates outcomes with universal consistency, the "social law" dictating that, for instance, a specific promise (social action A) leads to increased trust (social state B) is contingent upon cultural norms, prior relationship history, the perceived sincerity of the promiser, and numerous other evolving contextual factors. Its predictive power is not fixed but probabilistic and adaptive. Integrated world models must therefore embody the principle of capturing this contingent causality inherent in social dynamics. This demands capabilities to: (a) model how the causal pathways linking social antecedents to social and physical consequences can evolve, strengthen, or weaken; (b) represent how heterogeneous agents might understand and enact different causal models of their social world, leading to divergent behavioral patterns even in similar physical settings; and (c) ensure that the model's predictions of social behavior reflect this understanding of conditional, rather than deterministic, causality, moving beyond simplistic rule-following to a nuanced appreciation of strategic interaction and socially constructed realities.

**Entangled System Emergence and Co-evolution.** Ultimately, physical and social dimensions constitute a deeply entangled, co-evolving, holistic system where properties of the whole often transcend the sum of its parts. Actions and changes within one domain invariably and causally influence the other, creating complex feedback loops that drive the system's overall trajectory and give rise to emergent phenomena. This principle calls for a unified modeling approach that explicitly enables the simulation of this holistic system co-evolution and the emergence of novel, system-level properties resulting from physical-social entanglement. This entails capabilities to: (a) represent how physical environmental affordances causally shape social cognition and normative structures; (b) conversely, model how social states drive physical actions that modify the material world, capturing bidirectional causality; and (c) design architectures and learning paradigms that foster the emergence of complex collective behaviors and societal-level transformations from the interplay of numerous agents operating under both physical laws and multifaceted social influences.

These **ACE Principles** are not merely additive but deeply synergistic. Effective Abstraction is a prerequisite for understanding the Contingent Causality inherent in social dynamics. Both, in turn, are crucial for modeling the profound Entanglement between physical and social systems from which co-evolution and holistic emergence arise. Together, they form a cohesive set of guidelines for advancing towards truly comprehensive and predictive physical-social world models.

### 3.3 Framework for Unified Physical-Social World Model

Building upon the imperative for integration and the ACE principles, we propose a conceptual framework for unified physical-social world models (WM$_{\text{P-S}}$). Visualized in Figure 3, this framework emphasizes a principled approach to synergistically modeling and predicting the distinct yet deeply entangled dynamics of the physical and social realms. Our primary focus is on the **predictive problem**: learning and forecasting state transitions within complex environments comprising both physical and social elements. This problem is formulated as WM$_{\text{P-S}} = \langle N, \mathcal{S}, T \rangle$, where $N = \{1, \dots, n\}$ be the set of agents. The world state space $\mathcal{S}$ is a composite of physical and social dimensions:

$$\mathcal{S} = \mathcal{S}_{\text{phy}}^{\text{env}} \times \left( \times_{i=1}^{n} \mathcal{S}_{\text{phy}}^{i} \right) \times \left( \times_{i=1}^{n} \mathcal{S}_{\text{soc}}^{i} \right) \times \left( \times_{i,j \in N, i \neq j} \mathcal{S}_{\text{soc}}^{ij} \right).$$

Specifically, a specific world state $\mathbf{s} \in \mathcal{S}$ explicitly decomposes into a **joint physical state** $\mathbf{s}_{\text{phy}} = (s_{\text{phy}}^{\text{env}}, \{s_{\text{phy}}^{i}\}_{i \in N})$, where $s_{\text{phy}}^{\text{env}}$ describes the environment and $s_{\text{phy}}^{i}$ is agent $i$'s physical state, and a **joint social state** $\mathbf{s}_{\text{soc}} = (\{s_{\text{soc}}^{i}\}_{i \in N}, \{s_{\text{soc}}^{ij}\}_{i,j \in N, i \neq j})$, where $s_{\text{soc}}^{i}$ denotes individual social attributes (e.g., beliefs, goals) and $s_{\text{soc}}^{ij}$ captures inter-agent social relationships, the construct of $\mathbf{s}_{\text{soc}}$ follows the Principle A. Given the current world state $\mathbf{s} = (\mathbf{s}_{\text{phy}}, \mathbf{s}_{\text{soc}})$ (and potentially a joint action $\mathbf{a}$ if explicitly modeled), the core predictive challenge is to learn the **joint state transition function** $T(\mathbf{s}'|\mathbf{s})$, predicting the next state $\mathbf{s}'$. It is crucial to understand that this unified $T$ is not merely an additive combination of independent physical ($T_{\text{phy}}$) and social ($T_{\text{soc}}$) transition functions. Such isolated learning would fail to capture their deep coupling and reciprocal influence, as emphasized by Principle E. The evolution of $\mathbf{s}_{\text{phy}}$ is continuously affected by $\mathbf{s}_{\text{soc}}$ and vice-versa. Therefore, $T$ must inherently model this entanglement. Our proposed WM$_{\text{P-S}}$ framework, depicted in Figure 3,

operationalizes this unification. The learning, structure, and predictive mechanisms of this WM$_{\text{P-S}}$ are fundamentally guided by the overarching **ACE Principles**. Principle A shapes the multi-level abstraction of $\mathbf{s}_{\text{soc}}$; Principle C ensures the social component of predictions reflects contingent causality sensitive to the socio-physical context; and Principle E mandates the model captures the entangled co-evolution leading to holistic system emergence. The successful instantiation of such a WM$_{\text{P-S}}$, is envisioned to unlock a new generation of AI capabilities. As highlighted in Figure 3, this ranges from developing Smart Urban Mobility systems that understand both traffic physics and human driver behavior, to fostering truly Human-AI Teaming through mutual understanding, enabling more effective Government Policy Simulation, creating deeply engaging Advanced Game AI, supporting Personalized Well-being applications that consider socio-physical contexts, and improving Crisis Response by modeling human behavior under duress within physical constraints. This framework, therefore, not only addresses the limitations of current models but also charts a path towards AI that can more comprehensively understand and interact with our multifaceted world.

## 4 Challenges and future research directions

Scaling the ACE principles to real-world applications encounters significant barriers, each tied to fundamental AI challenges, yet these can be addressed through strategic development paths that leverage interdisciplinary insights, advanced computational methods, and innovative data strategies. We briefly discuss these challenges and propose several research directions to inspire future research.

### 4.1 Challenges in Scaling the Abstraction Principle

The Abstraction principle grounds abstract social concepts (e.g., "trust") in multimodal data without oversimplification, facing the neural-symbolic grounding problem [40]. Mathematically, learn $f : \mathcal{X} \rightarrow \mathcal{S}_{\text{soc}}$, where $\mathcal{X}$ is multimodal inputs (e.g., video $x_v$, audio $x_a$, text $x_t$), and $\mathcal{S}_{\text{soc}}$ is high-dimensional and sparse, complicating loss minimization $\mathcal{L}(f(x), s_{\text{true}})$ via cross-entropy or contrastive objectives. The core challenge lies in bridging the semantic gap from continuous, high-dimensional perceptual data (e.g., complex micro-expressions and body language in a video) to discrete, symbolic social concepts (e.g., "intention is cooperative"). This representation issue arises because social quantities lack the dimensional clarity of physical quantities, leading to ambiguity in encoding abstract notions into structured forms. Additional challenges include: (1) data-related limitations, such as scarcity of diverse multimodal datasets and overfitting to cultural or environmental biases, which hinder the abstraction process by limiting the breadth and fairness of learned mappings from perceptual inputs to symbolic outputs [58](2) computational and integration hurdles, including inefficiency in bridging sensory-symbolic gaps and the need to align with physical priors, directly impacting the scalability and grounding of abstractions in real-world, 3D-constrained environments; and (3) ethical and adaptability concerns, including the amplification of gaps due to underrepresented behavioral biases and the challenges of extending to lifelong learning [79, 111, 73], undermine the ethical integrity and continuous evolution of abstracted social representations.

### 4.2 Challenges in Scaling the Contingent Causality Principle

This principle handles non-stationary social rules via state transitions $P(s'_{\text{soc}}|s_{\text{soc}})$. The core difficulty lies in predicting social state changes after establishing representations, as social quantities evolve based on dynamic, contingent contexts and scenarios, unlike physical quantities governed by fixed laws. This contingency implies weak Markovian properties, where long-term dependencies and events with uncertain timing can influence state transitions, further constrained by evolving social norms. This leads to high out-of-distribution (OOD) variance $\text{Var}[P(s'_{\text{soc}}|c_{\text{OOD}})] \gg \text{Var}[P(s'_{\text{soc}}|c_{\text{in}})]$, where $c$ denotes the dynamic context (e.g., cultural or situational factors), requiring models to adapt predictions to shifting causal rules [95]. Additional challenges include causal multiplicity and uncertainty management, where models struggle to simultaneously handle multiple, coexisting causal rule sets (e.g., conflicting cultural norms in a single scenario) and manage uncertainty in state transitions due to incomplete or ambiguous contextual cues, complicating accurate and robust prediction of social outcomes in dynamic, norm-driven environments.

### 4.3 Challenges in Scaling the Entangled Emergence Principle

This requires modeling bidirectional loops in joint transitions $T(s'_{\text{phy}}, s'_{\text{soc}}|s_{\text{phy}}, s_{\text{soc}})$, with entanglement $I(s_{\text{phy}}; s_{\text{soc}}) > 0$. The core challenge is capturing mutual influences between physical

and social dimensions, where interactions lead to emergent behaviors unpredictable from isolated components—unlike separable physical systems, social-physical entanglements amplify complexity through feedback loops [43, 17]. State explosion and chaos (e.g., Lyapunov $\lambda > 0$, amplification $\Delta s_{t+1} \approx e^{\lambda \Delta t} \Delta s_t$) exacerbate this [110]. Additional challenges include:(1) cascading errors and systemic fragility, where accumulated errors over long time horizons and sensitivity to exogenous noise propagate through entangled states, destabilizing the system due to its inherent feedback loops and interconnectedness, severely undermining the reliability of predictions in socio-physical systems, as evidenced by nonlinear error amplification across interfaces [12] (2) unconstrained emergent complexity and unidentified socio-physical norms, encompassing the fundamental issue of inadequate spatial-physical grounding to constrain emergent interactions, risking unrealistic entanglements, and the substantial data requirements for accurately learning mutual influences that drive collective emergent patterns under evolving or unrecognized socio-physical norms, highlighting the difficulty in uncovering and applying implicit rules governing co-evolutionary dynamics (3) irreducible modeling complexity and unpredictable global impacts, where the intrinsic nonlinearity and high dimensionality of socio-physical entanglement render effective simplification or decomposition profoundly challenging, termed "irreducible" as core behaviors and emergent patterns reside in the continuous interplay between dimensions which fundamentally limiting the ability to predict or steer large-scale global phenomena, with systemic consequences emerging from complex interactions often remaining opaque and difficult to manage, including ethical oversight [42].

### 4.4 Future research directions

Developing truly unified physical-social world models based on the ACE Principles will necessitate concerted efforts across several key research directions. Future work on **Data Foundations** could focus on cultivating rich and diverse multimodal datasets, potentially through large-scale curation, augmentation, and synthetic generation. Such data would be crucial for establishing robust social abstractions, facilitating contingent predictions, and enabling dynamic entangled simulations, perhaps via hybrid neuro-symbolic methods or pre-training on diverse social interaction data. In **Architecture Design**, research might explore hybrid neuro-symbolic systems with modular components to enable grounded, bidirectional interactions between physical and social representations. This approach could address structural integration and entanglement challenges by incorporating inductive biases from cognitive science and behavioral economics, thereby informing models about context-dependent human decision-making and group behaviors. For **Algorithm Optimization**, advanced learning and reasoning methods will likely be essential, tailored for dynamic updates and uncertainty handling. These could enhance causal inference robustness and emergent behavior resilience through techniques such as meta-learning for non-stationary social rules, or systems theory-inspired approaches that combine multi-agent reinforcement learning, hierarchical abstractions, and physics simulators to model bidirectional physical-social feedback loops. Finally, **Evaluation and Scaling** should advocate for multi-tier performance metrics that assess both physical and social aspects, including entangled socio-physical causality. Scaling mechanisms like federated learning and ethical audits will be vital to ensure lifelong adaptation in dynamic, real-world environments and to mitigate biases inherent in complex socio-physical interactions.

## 5 Conclusion

This position paper has championed a pivotal paradigm shift for AI world models: the deep, bidirectional unification of their physical and social predictive capabilities. We argued that the prevalent separation of these dimensions renders current models fundamentally incomplete, hindering their capacity to capture the true complexity of our entangled physical-social reality and impeding progress towards AI systems that genuinely comprehend our multifaceted world. To chart a constructive path forward, we delineated the distinct natures of physical and social predictions, underscored the imperative for their integration by highlighting their reciprocal interplay, critiqued existing formulations, and subsequently introduced three foundational **ACE Principles**. These principles, together with our proposed conceptual WM$_{\text{P-S}}$ framework and a research roadmap, collectively offer a structured, principled approach to developing world models that holistically represent and predict the co-evolution of physical and social realities.

## Acknowledgments and Disclosure of Funding

This work is supported by the National Science and Technology Major Project (No. 2022ZD0114904).

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

## A    Illustrative Case Study: Service Robots in Eldercare

To illustrate why world models must jointly represent physical and social dynamics, consider a service robot assisting older adults in an eldercare environment. A subtle physical signal, such as a hand tremor detected in the sensory stream $s_{phy}$, is not merely a motor irregularity. It often corresponds to a latent social or emotional state, such as anxiety or unease, denoted as $s_{soc}$. This relationship is deeply bidirectional. The person's internal state influences physical outcomes: heightened anxiety increases the likelihood of dropping objects or moving unpredictably. Conversely, physical factors such as a cluttered room, sudden noise, or the robot's abrupt movement can heighten stress or discomfort. These reciprocal effects create a continuous feedback loop, where social and physical processes shape each other over time. A world model that encodes only physical dynamics can predict trajectories and collisions, but it will miss the human causes behind them. A model limited to social reasoning may infer anxiety but fail to anticipate its physical consequences. Only a unified world model, capable of representing the intertwined causal structure between $s_{phy}$ and $s_{soc}$, can anticipate how social states alter physical events and vice versa.

This case highlights that understanding and predicting human-centered environments requires more than physical simulation or social inference in isolation. Effective intelligence depends on capturing their entanglement within a single, coherent world model that links perception, causality, and interaction across both domains.

## B    Detailed World model methods integration table

This appendix section presents a detailed tabular survey of world model methodologies (Table 1 and Table 2), split into Physical World Models and Social World Models for clarity, as discussed in section 2. The tables provide comprehensive details including the *Characteristic* (e.g., explicit/latent for physical representation style, individual/interaction/group for social interaction level), *Architecture* (e.g., Transformer), *Task* (e.g., video generation), and a *Brief Description*. Citation numbers are added next to each algorithm name for reference. This integration highlights the strengths of existing methods in modeling physical dynamics (e.g., objective laws in MBRL) and social dynamics (e.g., agent interactions in LLM agents), while also revealing gaps in unification, such as the lack of entangled socio-physical representations. Readers can use these tables to identify opportunities for hybrid approaches that bridge the physical-social divide, consistent with the ACE Principles proposed in subsection 3.2.

The table columns are as follows: *Algorithm* (method name with citation), *Characteristic* (indicating model representation style for physical or interaction level for social), *Architecture* (e.g., Transformer), *Task* (e.g., Video Generation), and *Brief Description*. Category headers use lighter background colors inspired by Figure 2 (light blue for physical, light orange for social) for visual distinction and academic tone.

## C    Hierarchical Evaluation Protocol

To systematically assess a world model's ability to capture socio-physical dynamics, we propose a three-tier hierarchical evaluation protocol. Table 3 summarizes each tier from low-level perception to high-level causal integration. Overall, these tiers form a compact and principled evaluation framework that may help unify assessments of perceptual fidelity, modular reasoning, and causal entanglement within a single testing paradigm.

Overall, these tiers form a compact and principled evaluation framework that may help unify assessments of perceptual fidelity, modular reasoning, and causal entanglement within a single testing paradigm.

Table 1: **Physical World Model Methods Integration Table.** Comprehensive classification of physical methods with additional details.

| Algorithm | Explicit | Latent | Architecture | Task | Brief Description |
|---|---|---|---|---|---|
| **Intuitive Physics** | | | | | |
| MAC [48] | ✓ | | Recurrent Attention Network | CLEVR | Explicit multi-step visual reasoning via recurrent attention and control units. |
| DCL [15] | ✓ | | Propagation Network | CLEVRER | Grounds concepts from video and language. |
| Self-Supervised Intuitive Physics [31] | | ✓ | JEPA | Masked Region Prediction | Emerges intuitive physics from natural videos. |
| DINO as Representations [7] | | ✓ | Transformer | Intuitive Physics Benchmarks | Uses DINO for video world models in benchmarks. |
| V-JEPA 2 [4] | | ✓ | Transformer | Video prediction | Predictive latent model for intuitive physical understanding. |
| Cosmos-Reason1 [5] | | ✓ | Foundation Model | Physical Reasoning | Generates embodied decisions for physical understanding. |
| Cosmos World Platform [1] | | ✓ | Platform | Physical AI Setups | Platform for customized physical world models. |
| **MBRL** | | | | | |
| MBPO [49] | ✓ | | Ensemble MLP | MuJoCo | Improves RL sample efficiency with rollouts. |
| MOPO [124] | ✓ | | Ensemble MLP | D4RL | Safe offline RL with model uncertainty penalties. |
| Dreamer [34] | | ✓ | RSSM | Video Game | Learns behaviors via latent imagination. |
| Day-Dreamer [115] | | ✓ | RSSM | Rototic | Applies world models to physical robots. |
| TDMPC [38, 37] | | ✓ | TOLD | DMControl | Temporal difference for model predictive control. |
| IRIS [69] | | ✓ | Transformer | Atari | Sample-efficient world models for RL. |
| Transformer World Models [18] | | ✓ | Transformer | Craftax-Classic | Data-efficient RL with transformers. |
| R2I [90] | | ✓ | S4 | Long-horizon tasks | Improve long-term memory and credit assignment. |
| S4WM [21] | | ✓ | S4 | Long-horizon tasks | Improve stability and sample efficiency in long-horizon tasks. |
| MuZero [96] | ✓ | ✓ | Recurrent dynamics + MCTS planner | Atari / Go / MuJoCo | Combines latent dynamics learning with tree search planning. |
| **Video Generator** | | | | | |
| Stable Video Diffusion [10] | | ✓ | Latent Diffusion Model | Video Generate | High-resolution text-to-video generation. |
| DynamiCrafter [121] | | ✓ | Latent Diffusion Model | Video Generate | Animates images with diffusion priors. |
| Open-Sora [63] | | ✓ | Latent Video Diffusion Transformer | Video Generate | Open-source large video generation model. |
| Video World Models Memory [116] | | ✓ | Spatial Memory-Augmented Transformer | Long-Horizon Consistency | Enhances consistency with spatial memory. |
| Interactive Video Generation [14] | | ✓ | Action-Conditioned Transformer | Video Planning | Learns interactive video with coherence. |
| Pandora [118] | | ✓ | Autoregressive-Diffusion Video Model | Interactive Video Generation | Generate videos from natural-language actions. |
| Navigation World Models [9] | | ✓ | Conditioned Video Transformer | 3D Navigation | Controllable videos for navigation tasks. |
| HunyuanWorld [109] | | ✓ | Transformer + Diffusion hybrid | Video understanding and generation | Large unified video world model |
| **LLM for Physical** | | | | | |
| RAP [39] | | ✓ | Transformer + MCTS | Math & Logical | Reasoning with planning and world models. |
| WorldCoder [107] | | ✓ | Transformer+Program-Synthesized World Model | AlfWorld | Builds world models via code generation. |
| CWMI [97] | ✓ | | Transformer+Causal Physics Module | Zero-Shot Physical Reasoning | Induces causal world models in LLMs. |
| World Knowledge Model [85] | | ✓ | Transformer+Parametric World Knowledge Model | Interactive Agent Planning | Provides prior task knowledge to assist agent planning. |
| **3D World Model** | | | | | |
| OccWorld [133] | ✓ | | Spatial-Temporal Generative Transformer | 3D Occupancy Prediction | 3D occupancy for autonomous driving. |
| 3D Persistent World Models [134] | | ✓ | Transformer+Persistent Memory Module | Long-Horizon 3D Generation | Consistent 3D embodied models. |
| Matrix-3D [122] | | ✓ | Video Diffusion+3D Reconstruction | 3D World Generation | Omnidirectional 3D world generation. |
| Gaussian World Model [136] | ✓ | | 3D Gaussian representation | 3D Occupancy Prediction | Streaming 3D occupancy prediction. |
| GWM [65] | | ✓ | 3D VAE +DiT | Robotic Manipulation | Scalable World Models for Robotic Manipulation |

Table 2: **Social World Model Methods Integration Table.** Comprehensive classification of social methods with additional details.

| Algorithm | Individual | Interaction | Group | Architecture | Task | Brief Description |
|---|---|---|---|---|---|---|
| **ToM** | | | | | | |
| ToMnet [86] | ✓ | | | LSTM | Sally-Anne test | Predicts agent behaviors with ToM. |
| M³RL [100] | ✓ | ✓ | | LSTM | Management | Mind-aware multi-agent coordination. |
| ToM Goes Deeper [113] | ✓ | | | LLM | ToM Capabilities | Investigates deeper ToM capabilities. |
| Decompose-ToM [93] | ✓ | | | LLM | ToM Reasoning | Decomposes ToM tasks for reasoning. |
| Discrete World Models [45] | ✓ | | | LLM | ToM Reasoning | Measures task difficulty via structured ToM reasoning. |
| DynToM Mental State Alignment [119] | ✓ | ✓ | | LLM | Dynamic ToM Alignment | Predictive social interaction in world models. |
| **MBMARL** | | | | | | |
| Networked MBMARL [66] | | ✓ | ✓ | GNN+MLP | CACC | Efficient MARL for large-scale network control. |
| MAG [117] | | ✓ | | RSSM | SMAC | Models agents for strategic games. |
| Sequential World Models [132] | | ✓ | | Sequential agent-wise world models | Multi-Robot Cooperation | Enhances multi-robot cooperation. |
| Global-Aware World Model [99] | | ✓ | | Transformer | SMAC | Unified representations in MARL. |
| DIMA [130] | | ✓ | | Diffusion | SMAC+MPE | Diffusion-inspired state space model. |
| Decentralized Transformers [129] | | ✓ | | Transformer | SAMC | Decentralized transformers for MARL. |
| SWM-AP[127] | ✓ | | ✓ | MLP+LSTM+GNN | Mechanism Design | Counterfactual world model for mechanism design. |
| **LLM for Social** | | | | | | |
| Social Alignment[64] | ✓ | ✓ | | Single LLM | Social Feedback Alignment | Predict social value dynamics in world modeling. |
| Cultural Value Alignment Eval [104] | ✓ | | ✓ | Single LLM | Cultural Preference Alignment | Social norm prediction in dynamic environments. |
| Strong-Weak Value Alignment [54] | ✓ | ✓ | | Single LLM | Human Value Alignment | Social decision-making and mental states. |
| Generative Agents [74] | | | ✓ | LLM Agents | Social Simulation | Simulates human-like behaviors. |
| Evobot [56] | | | ✓ | LLM Agents+GNN | Social simulation | Generate more human-like content |
| SocioVerse [128] | | | ✓ | LLM Agents | Social Simulation | LLM-driven world model with alignment. |

# D   Current benchmarks table

This section extends the analysis in section 2 by providing a comparative overview of current world model benchmarks (Table 4). As illustrated in the table, existing benchmarks offer valuable coverage of either physical or social dynamics, yet they seldom capture the intertwined nature of socio-physical causality. The evaluation framework outlined here examines each benchmark's level of support for physical reasoning, social reasoning, and, critically, entangled socio-physical interactions, while summarizing their primary characteristics in a concise *Brief Description* column. The comparison reveals several structural gaps across current benchmarks. Physically grounded benchmarks (e.g., MuJoCo) achieve strong performance in objective simulation but largely omit social interaction aspects, whereas socially focused benchmarks (e.g., SocialIQA) often lack explicit

Table 3: Three-tier hierarchical evaluation protocol for assessing socio-physical dynamics in world models. Each tier targets an increasingly integrated understanding of physical and social dynamics.

| Tier | Core Objective & Example Tasks | What It Verifies |
|---|---|---|
| **Perceptual Fidelity** | *Objective:* Assess low-level sensory prediction across modalities (visual, textual, embodied). *Examples:* Reconstruct video frames, facial expressions, or gestures. | Ensures perceptual grounding and multimodal fidelity without distortion. |
| **Disentangled Dynamics** | *Objective:* Evaluate physical and social reasoning independently before integration. *Examples:* Simulate trajectories under gravity; infer isolated intentions or preference alignment. | Verifies modular reasoning and the model's compliance with underlying physical laws and social norms prior to integration. |
| **Entangled Dynamics** | *Objective:* Probe bidirectional socio-physical causality via counterfactuals. *Examples:* "If anxiety rises in a crowd, how does movement change?" or "If cooperation norms shift mid-interaction, what are the physical effects?" | Validates causal interplay and generalization in entangled socio-physical contexts. |

Table 4: Comparative overview of current world model benchmarks and their evaluation coverage, organized by physical-, social-, and unified-focus categories. The *Brief Description* column summarizes each benchmark's main characteristics.

| Benchmark | Year | Focus | Brief Description |
|---|---|---|---|
| **Physical-Focused Benchmarks** | | | |
| MuJoCo [112] | 2012 | Physics Simulation | Standard physics engine for control and dynamics evaluation. |
| CLEVR [50, 123] | 2017 | Visual Reasoning | Synthetic visual reasoning benchmark for compositional scenes. |
| CARLA [23] | 2017 | Autonomous Driving | High-fidelity driving simulator for embodied decision-making. |
| DMControl [108] | 2018 | Control Tasks | Continuous-control suite for reinforcement learning evaluation. |
| Habitat [94] | 2019 | Embodied AI | 3D embodied navigation platform with realistic rendering. |
| D4RL [29] | 2020 | Offline RL Datasets | Standard offline RL datasets for policy and model evaluation. |
| MineDojo [25] | 2022 | Minecraft Tasks | Open-ended platform emphasizing embodied physical interaction. |
| IntPhys [89, 11] | 2018 | Physical Reasoning | Visual intuitive-physics benchmark for physical consistency. |
| CausalVQA [28] | 2025 | Physical Reasoning | Video-based benchmark for latent physical reasoning. |
| WorldModelBench [61] | 2025 | Video World Models | Unified evaluation for generative and predictive video models. |
| **Social-Focused Benchmarks** | | | |
| Sally-Anne Test [86] | 2018 | Theory of Mind | Classic ToM paradigm for belief and perspective inference. |
| Stag Hunt Game [62] | 2019 | Cooperative Games | Coordination game modeling social dilemmas and cooperation. |
| SocialIQA [92] | 2019 | Social Commonsense | QA benchmark for intentions and social reasoning. |
| ATOMIC [91] | 2019 | Commonsense Knowledge | Knowledge graph for causal and social event reasoning. |
| MMMU [125] | 2024 | Multimodal QA | Multimodal academic QA benchmark; non-social reasoning only. |
| MuMA-ToM [98] | 2025 | Multimodal ToM | Multimodal ToM benchmark combining visual and textual cues. |
| Egonormia [87] | 2025 | Social Norms | Tests norm understanding in embodied social contexts. |
| UserBench [84] | 2025 | User-Centric Agents | Evaluates user-aligned adaptation in interactive settings. |
| HumanTrait [41] | 2025 | Personality Modeling | Studies personality-based reasoning and social adaptation. |
| **Unified Benchmarks** | | | |
| Overcooked [13] | 2019 | Kitchen Cooperation | Cooperative cooking under spatial and social constraints. |
| Neural MMO [103] | 2019 | Massively Multiplayer | Persistent multi-agent world for emergent social behavior. |
| Melting Pot [60] | 2021 | Multi-Agent Dilemmas | Benchmark suite for adaptive cooperation and competition. |
| CivRealm [83] | 2024 | Adaptive Social | Civilization-style simulation for strategic and social learning. |
| AdaSociety [47] | 2024 | Adaptive Social | Adaptive multi-agent society with evolving interactions. |
| ProjSId [2] | 2024 | Minecraft Social Simulation | Combining physical causality with emergent social behaviors. |

physical grounding. Some integrated environments, such as Melting Pot, show promise for joint evaluation but remain limited in counterfactual testing and long-term causal entanglement. Overall, this comparative analysis highlights the need for more comprehensive and unified evaluation protocols that can systematically assess the full spectrum of physical, social, and socio-physical reasoning, aligning with the broader research directions discussed in section 4. Moreover, this overview aims to encourage future benchmark development guided by the proposed *Hierarchical Evaluation Protocol* (Table 3), fostering more structured and hierarchical evaluation of world models.

# E   Comparison with Existing Positions and Surveys on World Models

Existing surveys on world models primarily provide technical overviews of predictive capabilities, multimodal integration, or specific domains like embodied AI or 3D modeling, often neglecting the bidirectional unification of physical and social dynamics. In contrast, our position paper adopts a novel dual-lens perspective, framing world models through physical and social dimensions, highlighting their entanglement. We draw inspiration from cognitive science, sociology, and systems theory to construct holistic models, and we provide paths and evaluations to inspire future work. The following table summarizes comparisons with selected recent surveys and position papers, chosen for their recency and relevance.

Table 5: Comparison with Existing Positions and Surveys on World Models

| Survey/Position | Main Focus |
|---|---|
| Autonomous Machine Intelligence [59] | JEPA |
| Multimodal WM [67] | Transition from multimodal to world models |
| Autonomous Driving WM [32] [27] | World models for autonomous driving |
| Edge AI WM [131] | Edge intelligence and agentic AI via world models |
| WM Overview [22] | Perspective of Prediction and Understanding |
| Sensing, Learning, Reasoning [20] | World models for sensing, learning, and reasoning in AI |
| Embodied AI WM [26] | Embodied AI: From LLMs to world models |
| Generative WM [135] | Generative world models and simulators (e.g., Sora) |
| MBRL Survey [71] | Model-based reinforcement learning |
| WM Critiques [120] | Physical, Agentic, and Nested |
| General Agents WM [88] | Shows general agents contain world models |
| Video Gen WM Perspective [53] | Physical law perspective on video generation as world models |
| Phys Interpretable WM [77] | Four principles for physically interpretable world models |
| 3D/4D WM Survey [57] | Survey on 3D and 4D world modeling |
| LLM Social Sim [106][3] | Integrating LLMs in agent-based social simulation |
| Modeling the World [30] | Embodied agents' world modeling |

