# OpenReview forum: "World Models Should Prioritize the Unification of Physical and Social Dynamics"
_NeurIPS.cc/2025/Position_Paper_Track — NeurIPS 2025 Position Paper Track_

### Official Review · Reviewer_xcmG · 2025-07-11

**Significance:** 3
**Presentation:** 3
**Rating:** 6
**Confidence:** 4

**Summary:**

- this paper proposes theoretical guidelines/principles on designing intelligent systems that integrates physical and social dynamics, rather than building world models that consider only one of the two dimensions. The principles are briefly discussed and a framework definition is proposed that creates joint physical and social states.

**Strengths:**

- it tackles a well-known limitation of current intelligent agents' world models which is of interest of the NeurIPS community
- the proposed framework is clearly presented and supported by most up-to-date related approaches
- it provides a long list categorizing existing world model methods into physical and social

**Weaknesses:**

- this work does not mention digital twins which increasingly have been modelling physical and social dimensions together
- limitations of existing approaches are not discussed in depth (simulated environments with traditional text-based LLMs)
- more discussion on multimodality and its limitations compared to the proposed framework

**Questions:**

- which are the current benchmarks or tasks that could show that current models fail to integrate social-physical aspects?
- the example provided: "A purely physical model might predict a ball’s trajectory if thrown, but ... its physical characteristics." -> If context (video) is provided to a multimodal model, won't it be able to say the intent of throwing a ball?
- how does your proposal compares to social digital twins? (https://www.mdpi.com/2624-6511/8/1/23)
- there is some work on multiagent environment that integrates social and physical aspects (https://arxiv.org/abs/2506.12331v1) and also
- project Sid hasn't been discussed where multi agent Minecraft simulations with thousands of agents simulate physical and social interactions to build civilizations (https://arxiv.org/html/2411.00114v1). Despite not having joint states, what are the limitations of these simulated environments?
- physical and social dimensions are not already implicitly embedded in multimodal systems such as Veo3 or even OpenAI o3? What would be the advantage of creating physical-social states instead of simply use multimodality? Aren't those states already capturing physical and social dynamics (among other dimensions)?

**Alternative Position:**

Yes, and alternative positions are well-considered and named but not addressed

**Author Identification:**

No.

**Context:**

3

**Details Of Ethics Concerns:**

No concern.

**Discussion:**

4

**Ethics:**

["NO or VERY MINOR ethics concerns only"]

**Position:**

Yes, the paper argues for or against a position related to machine learning.

**Support:**

3

**Thoroughness:**

4

---

### Official Review · Reviewer_Do99 · 2025-07-14

**Significance:** 3
**Presentation:** 2
**Rating:** 6
**Confidence:** 3

**Summary:**

This position paper argues that the next major step for world model development is the bidirectional unification of physical and social dynamics. Current models are siloed, either on physical prediction (e.g., reinforcement learning, intuitive physics) or social modeling (e.g., preferences, Theory of Mind), which limits their applicability in complex real-world scenarios where these dimensions interact. The paper reviews the limitations of both paradigms, proposes the ACE Principles (Abstraction, Contingent causality, Entangled emergence) for guiding integrated modeling, and introduces a conceptual framework (WMP-S) that treats the physical-social world state as a coupled system. The authors advocate for this unified modeling as essential for building robust, generalizable AI systems capable of interacting meaningfully in human-centric environments.

**Strengths:**

The paper presents a novel, forward-looking position rooted in theoretical rigor and empirical insight. It provides a comprehensive analysis of the limitations of current physical and social world models, synthesizes interdisciplinary knowledge, and proposes actionable principles (ACE) for future research. Its conceptual framework (WMP-S) offers a clear path forward. The writing is clear, the motivation is well-articulated, and the topic is of high relevance to NeurIPS and broader AI research.

**Weaknesses:**

The proposal is largely conceptual and could be further strengthened by empirical case studies or concrete prototype implementations. While the unification argument is compelling, more detail on how to resolve practical issues like data alignment, computational scalability, and evaluation of integrated models would be valuable. The ACE Principles are sound but might benefit from comparative examples or counterpoints. There is also limited discussion of how this integration might be applied across diverse cultural or ethical contexts.

**Questions:**

- How might the proposed WMP-S framework be evaluated in practice? Are there benchmark environments where physical and social dynamics are sufficiently represented?
- What are the most critical technical barriers to implementing the ACE principles at scale?
- How might this framework account for culture-specific social dynamics that don’t generalize easily?

**Alternative Position:**

Yes, and alternative positions are well-considered and addressed by the argument

**Author Identification:**

No.

**Context:**

3

**Discussion:**

3

**Ethics:**

["NO or VERY MINOR ethics concerns only"]

**Position:**

Yes, the paper argues for or against a position related to machine learning.

**Support:**

2

**Thoroughness:**

5

---

### Official Review · Reviewer_xq2p · 2025-07-22

**Significance:** 3
**Presentation:** 3
**Rating:** 7
**Confidence:** 4

**Summary:**

This position paper argues that the next major leap for AI world models lies in the deep, bidirectional unification of physical and social predictive capabilities. The authors point out that current research largely develops models for physical dynamics (e.g., model-based RL, video prediction) and social dynamics (e.g., preference modeling, Theory of Mind, LLM-based agents) in isolation, resulting in incomplete representations of real-world complexity. They analyze the limitations of this separation, survey the state of the art in both dimensions, and propose the ACE Principles as guiding tenets for future work: Abstraction of social complexity and heterogeneity, Capturing contingent causality, and Enabling entangled system emergence and co-evolution. They further provide a conceptual framework (WMP-S) and research roadmap for developing holistic world models that integrate both physical and social domains. The paper's central position is that such unification is essential for robust, generalizable, and socially-aware AI systems capable of navigating the full complexity of real-world environments.

**Strengths:**

1. Clear and timely position: The paper identifies a critical bottleneck in world model research and argues convincingly for its resolution.

2. Comprehensive literature review: Both physical and social world modeling paradigms are reviewed, making the gap and integration challenges concrete.

3. Conceptual innovation: The ACE Principles provide a useful intellectual framework that could guide future work.

**Weaknesses:**

1. Lack of concrete implementation or case studies: The framework is conceptual; no empirical demonstration or detailed implementation proposal is given, which may limit immediate practical uptake.

2. Evaluation and benchmarking: The paper does not propose specific evaluation metrics or benchmarks for integrated physical-social world models.

3. Scalability and data availability: While challenges such as data scarcity and complexity are acknowledged, more concrete strategies for addressing these (e.g., multi-modal data collection, transfer learning) would strengthen the roadmap.

**Questions:**

1. How might the field begin to build and benchmark the first integrated physical-social world models? Are there specific domains or simulation environments you recommend as testbeds?

2. Do you envision this integration as a monolithic architecture, or a modular combination of physical and social modules with shared representations?

**Alternative Position:**

Yes, and alternative positions are well-considered and named but not addressed

**Author Identification:**

No.

**Context:**

3

**Discussion:**

4

**Ethics:**

["NO or VERY MINOR ethics concerns only"]

**Position:**

No, the paper argues that a specific technical approach is superior to other approaches.

**Support:**

3

**Thoroughness:**

4

---

### Note · Authors · 2025-09-04

**1-11 Submit Again:**

Definitely yes

**1-1 Submission Process:**

5

**1-2 Next Year:**

We are strong proponents of the Position Paper track and greatly appreciate the platform it provides for discussing significant, forward-looking ideas. For next year, we would be excited to see the following enhancements:

**1-3 Future Development:**

Building on our positive experience with this track, we have a few ideas for its future development and improvement:
1. We suggest further emphasizing in the Call for Papers that the evaluation of position papers should prioritize the novelty of the position, the logical soundness of the argument, and the potential impact on future research, rather than demanding the exhaustive experiments or implementation details expected of a technical paper. This would help standardize the review process and keep the focus squarely on the "Position" itself.
2. We found the current author's survey/response process to be highly valuable. We hope this component can be leveraged more formally in the future. For instance, the final camera-ready version of an accepted paper could be accompanied by a curated summary of the author-reviewer discussion, allowing the wider community to appreciate the deeper context and dialogue surrounding the paper's position.
3. In addition to the open call, the track could introduce one or two "Thematic Challenges" each year, proposed by senior researchers in the field (e.g., "What are the fundamental limits of AI explainability?" or "What is the primary bottleneck for next-generation world models?"). This could focus the community's intellectual energy on critical, unresolved questions.

**1-4 Interest:**

["Panel discussions with other position paper authors", "Structured debates on controversial topics", "Workshops for developing position papers"]

**1-5 Thoughtful:**

9

**1-6 Supportive:**

9

**1-7 Technical Aspects Versus Position:**

7

**1-8 Gate Keeping:**

9

**1-9 Camera Ready Changes:**

1. As suggested by Reviewers xq2p and Do99, we will introduce a running case study of "Service Robots in Eldercare" to precisely illustrate our WMP-S framework. Here, physical cues like hand tremors signal latent social states like 'anxiety,' which causally predict future physical events (e.g., spilling a glass). Our framework demonstrates bidirectional entanglement by inferring anxiety and offering proactive assistance, rather than mere reactive clean-up, proving the necessity of unifying social and physical models for effective and empathetic care.

2. Responding to all reviewers' feedback on evaluation, we will add a new section. We will highlight current benchmarks' shortcomings and propose a novel, hierarchical evaluation protocol. This protocol comprises three tiers: Perceptual Fidelity, Disentangled Dynamics, and our key innovation, Entangled Dynamics. This final tier will use counterfactual probes to test the model's understanding of causal socio-physical interplay, transforming our critique into an actionable proposal for measuring progress.

3. We will articulate the fundamental advantage of our explicit, causal modeling over the implicit, correlational approach of large multimodal models, emphasizing its benefits for robust generalization and interpretability. Furthermore, we will clarify the synergy with concepts like Social Digital Twins and environments like Project Sid, positioning them as ideal platforms for training and testing our learning paradigm, rather than as alternative modeling approaches.

4. We will elaborate on technical barriers to scaling ACE principles, suggesting interdisciplinary pathways from cognitive science and behavioral economics. Crucially, we will explicitly address cultural specificity, explaining how our "Contingent Causality" principle models locally consistent social norms, fostering robust and ethically sound AI, particularly for culturally diverse contexts.

**3-1 Review Response1:**

xq2p

**3-2 Reaction To Review1:**

To ground our framework, we illustrate its necessity with a service robot case study in eldercare. A subtle physical cue, like a hand tremor, signals not merely a motor issue but a latent social state (e.g., anxiety). Our WMP-S framework will model how this inferred social state causally modulates the joint state transition function, predicting both physical risks (e.g., spilling a glass) and social needs (e.g., reassurance). This demonstrates that modeling social states is crucial for accurate physical prediction and effective care. Conversely, physical states (e.g., a cluttered room) can induce social states (e.g., stress), showcasing full bidirectional entanglement.

Our position also necessitates new evaluation metrics. We propose a hierarchical protocol: Tier 1, Perceptual Fidelity, for raw sensory data; Tier 2, Disentangled Dynamics, for isolated physical and social rules; and Tier 3, Entangled Dynamics. The final tier will use counterfactual probes to test the model's understanding of socio-physical causal interplay.

Regarding data, the core challenge is the separation of current sources, physics-rich video versus social-rich text. We argue for creating datasets that jointly capture entangled socio-physical interactions. Moreover, echoing Assessing Adaptive World Models in Machines with Novel Games (Tenenbaum et al., 2025), true world models must be refined through interaction and exploration. This necessitates developing interactive environments where agents can perform experiments to actively build their causal understanding through "world model induction."

Finally, on architecture, our position is conditioned on data and computing power. Currently, a principled modular approach is superior, allowing knowledge injection into specialized modules connected by a causal interface. In a future with massive, interactive datasets, a unified, end-to-end approach becomes compelling, allowing complex socio-physical dynamics to emerge.

**3-3 Review Response2:**

Do99

**3-4 Reaction To Review2:**

Regarding the weaknesses identified, we have developed a comprehensive plan to address them, detailed in our response to Reviewer xq2p, which includes a concrete case study, a new hierarchical evaluation protocol, and a discussion on creating new datasets.

Regarding the technical barriers to scaling ACE, these principles directly map to core difficulties in modern AI. Abstraction targets the fundamental neural-symbolic grounding problem: bridging the gap between raw sensory data, where neural models excel, and the abstract, symbolic concepts like 'trust' that govern social dynamics. Contingent Causality addresses the problem of distributional shift in non-stationary social rules, a notorious weakness of current ML models. We propose that by incorporating inductive biases from behavioral economics regarding human decision-making and interaction patterns, models can better understand and predict how social rules are adopted and enacted in specific contexts. This fundamentally enhances the model's ability to discern 'which set of social rules applies here,' moving beyond mere statistical fitting.". Finally, Entangled Emergence focuses on the immense challenge of modeling the reciprocal causal influence between physical and social domains; modeling this continuous, bidirectional feedback loop remains a major hurdle.

Regarding culture-specific social dynamics, our framework is explicitly designed to handle this via the Contingent Causality principle. Unlike a standard LLM that might average conflicting cultural norms, our WMP-S must learn a conditional model of social dynamics, inferring the active context and asking, "Which set of social rules applies here?" The goal is not a global average of human behavior, but to become adept at identifying and applying the locally consistent "social physics" of a given environment. This approach is a cornerstone of our proposal for building more robust and ethically sound AI systems.

**3-5 Review Response3:**

xcmG

**3-6 Reaction To Review3:**

Re: Implicit Modeling in Multimodal Systems (Q2, Q6): You ask if systems like Sora/Veo have already implicitly capture socio-physical dynamics. They do capture statistical correlations, allowing them to describe a social "intent" from video as a form of sophisticated pattern recognition. However, our argument is that this implicit approach is insufficient for robust AI. An explicit, causal representation is fundamentally superior because it is more sample-efficient by allowing the injection of known knowledge like physics and social norms, provides interpretability and controllability via explicit social states that act as "semantic handles," and enables robust generalization and cross-cultural adaptation by learning conditional rules rather than brittle, static patterns.

Re: Comparison to Other Environments (Q3, Q4, Q5): We appreciate your references to Social Digital Twins, Project Sid, and related works. We view them as highly complementary but serving different roles. These systems are typically bespoke simulators or environments with predefined dynamics, excellent at generating complex phenomena. Our WMP-S framework, conversely, is a general learning paradigm. Its goal is to enable an agent to learn the underlying generative model of such environments from observation. Thus, these systems are ideal testbeds for our proposed learning approach.

Re: Need for Benchmarks and Evidence of Failure (Q1): You ask for tasks demonstrating current models' failure. Failure is revealed in any task requiring an understanding of entangled socio-physical causality. For example, in our service robot scenario, a physical tremor directly signals a social state like 'anxiety.' An AI unable to make this inference will fail to provide appropriate care due to misunderstanding the physical world's social cause. Current benchmarks do not test this critical capability. In our revision, we will detail such examples and propose a new hierarchical evaluation protocol to fill this gap.

---

### Meta-Review · Area_Chair_Anvi · 2025-08-31

**Rating:** 7
**Confidence:** 4

**Strengths:**

This paper argues that progress in AI world models requires a more unified treatment of physical and social dynamics. The authors survey current work on each dimension and propose "ACE principles" plus a conceptual framework for modeling coupled physical-social states as a path towards more robust world models in AI.

According to the reviews, the paper had the following key strengths:
- The paper discusses a timely and important topic. It provides a thorough review and categorization of existing work studying world models. Consequently, the paper has a good shot to be a valuable reference for the field
- The paper makes a nice conceptual contribution in the ACE principles and WMP-S conceptual framework. This could provide a valuable roadmap for guiding future work.

**Weaknesses:**

The following are key weaknesses identified by the reviewers:

- The paper only provides an organization of existing work and conceptual frameworks. There is no empirical results nor case-studies, and so it is difficult to assess the practical upshot of the authors' argument.
- The paper provides no clear guidance on evaluation. How would we assess whether models have better integrated physical-social models in light of ACE principles and WMP-S?
- The paper should engage more with the possibility of multi-modal AI to address the gap between physical and social models.

**Questions:**

In addition to the main weaknesses identified above, each of the reviewers provide specific questions for the authors that would be valuable to address. I'd encourage the authors to particularly engage with these two:

- How might the field begin to build and benchmark the first integrated physical-social world models? Are there specific domains or simulation environments you recommend as testbeds? [Related: are examples like the minecraft simulation mentioned by one of the reviewers insufficient? If so, why?]
- What are the most critical technical barriers to implementing the ACE principles at scale?

**Ethics:**

The reviewers raised no ethical concerns about this paper.

**Thoroughness:**

2

---

### Decision · Program_Chairs · 2025-09-26

Accept